# Porous Anion Exchange Membrane for Effective Acid Recovery by Diffusion Dialysis

**Jinbei Yang [1] , Guangkai Dai [2], Jing Wang [3], Shuai Pan [3], Gang Lu [3], Xiaoke Shi [3], Danni Tang [3], Jinyi Chen [1] and Xiaocheng Lin [3,4,*]**

[1] Fujian Provincial Key Laboratory of Coastal Basin Environment, Fujian Polytechnic Normal University, Fuzhou 350300, China; yangjinbei@sina.com (J.Y.); jychen15@hotmail.com (J.C.)
[2] Department of Environmental Sciences, Faculty of Agriculture, Dalhousie University, Truro, NS B2N 5E3, Canada; gn527948@dal.ca
[3] College of Chemical Engineering, Fuzhou University, Fuzhou 350108, China; n180427063@fzu.edu.cn (J.W.); n190427054@fzu.edu.cn (S.P.); n190420042@fzu.edu.cn (G.L.); 041901430@fzu.edu.cn (X.S.); 041901422@fzu.edu.cn (D.T.)
[4] Fujian Science & Technology Innovation Laboratory for Chemical Engineering of China, Quanzhou 362114, China
[*] Correspondence: xclin@fzu.edu.cn

**Abstract:** Diffusion dialysis (DD) employing anion exchange membranes (AEMs) presents an attractive opportunity for acid recovery from acidic wastewater. However, challenges exist to make highly acid permeable AEMs due to their low acid dialysis coefficient ($U_{acid}$). Here, a series of porous and highly acid permeable AEMs fabricated based on chloromethyl polyethersulfone (CMPES) porous membrane substrate with crosslinking and quaternization treatments is reported. Such porous AEMs show high $U_{acid}$ because of the large free volume as well as the significantly reduced ion transport resistance relative to the dense AEMs. Compared with the commercial dense DF-120 AEM, our optimal porous AEM show simultaneous 466.7% higher $U_{acid}$ and 75.7% higher acid/salt separation factor ($S_{acid/salt}$) when applied to acid recovery at the same condition. Further, considering the simple and efficient fabrication process as well as the low cost, our membranes show great prospects for practical acid recovery from industrial acidic wastewater.

**Keywords:** anion exchange membrane; diffusion dialysis; acid recovery; porous membrane

## 1. Introduction

As industry developed, acidic wastewater pollution emerging from the industrial production in the metallurgical, steel pickling and surface cleaning processes has become a serious problem [1–5]. The conventional methods represented by neutralization, crystallization, thermal decomposition and solvent extraction are facing various problems including high energy consumption and further pollution by disposal of salts. By contrast, diffusion dialysis (DD) employing anion exchange membranes (AEMs) is a promising alternative for acid recovery from acidic wastewater. For example, a diffusion dialyzer using the commercial DF-120 AEM developed by Tianwei Membrane Co., Ltd., (Weifang, China) has achieved extensive industrialization [6]. As a spontaneous process merely driven by the concentration gradient, DD shows remarkable economic advantage, as compared with the above-mentioned conventional methods [6–8].

AEM is the core component of DD process for acid recovery, whose acid dialysis coefficient ($U_{acid}$) and acid/salt separation factor ($S_{acid/salt}$) during DD process are two decisive parameters for the evaluation of the acid recovery efficiency and the purity of the recovered acid, respectively. Until now, the large-scale application of DD is impeded by the major shortcoming of the unsatisfied DD performance especially the low $U_{acid}$ of the existing AEMs. For instance, the diffusion dialyzer equipped with 500 m² commercial DF-120 AEM can only treat 5.8 t acidic wastewater per day, due to the low $U_{acid}$ of the

DF-120 AEM. However, DD is still economically promising considering the increasing cost for the other common methods. Therefore, the improvement in $U_{acid}$ of AEMs is still greatly valuable to reduce operation cost and enhance competitiveness of DD.

It is noted that the common AEMs used in DD, represented by the commercial DF-120 AEM, are dense and symmetric in morphology. In recent years, the research interest of AEMs also mainly focused on the dense AEMs aiming for the improvement in the DD performance [9–19]. Emmanuel et al. [20] prepared the dense AEMs based on pyridinium functionalized 11-bromo-1-undecanol as the starting material, which was blended with tetraethylorthosilane (TEOS) and polyvinyl alcohol (PVA), followed by a sol-gel reaction. These membranes showed $U_{HCl}$ of 0.0174–0.0248 m h$^{-1}$ and $S_{HCl/FeCl_2}$ of 30.49–57.51 for the acid recovery from the HCl/FeCl$_2$ mixture solution at 25 °C. Irfan et al. [21] constructed the dense AEMs composed of PVA and pre-synthesized anion exchange precursor, which was synthesized through reaction of 1,5-diaminonaphthalene and 2,3-epoxypropyltrimethylammonium chloride, and the DD performance including 0.0194–0.025 m h$^{-1}$ $U_{HCl}$ and 27.84 to 52.60 $S_{HCl/FeCl_2}$ could be achieved for the obtained AEMs. In spite of the progress achieved by far, $U_{acid}$ of the dense AEMs are still unsatisfied. On the basis of the solution-diffusion mechanism, the transport rate of ions (i.e., H$^+$ and metal ions) across a dense or nano-porous polymer membrane is dictated to a large degree by the free space volume of the polymer [22,23], so the unsatisfied $U_{acid}$ of the dense AEMs should be attributed to a highly compact matrix resulting in the low free space volume.

In our previous work, various AEMs based on the porous membrane substrate (i.e., ultrafiltration membrane) had been fabricated for acid recovery by DD [24–26]. Different from the common dense AEMs, these porous AEMs were composed of a thin (1–2 μm) nano-porous selective layer and a macro-porous supporting layer, thus they showed much better DD performance, especially the higher $U_{acid}$ because of the higher free volume [22,23]. To further explore this new methodology for the porous AEM fabrication, it is of great importance to study the membrane formation and properties using a different polymer matrix. Until now, polyethersulfone (PES) as a common polymer for the porous membrane substrate has not been investigated for the porous AEM fabrication. In this study, chloromethyl polyethersulfone (CMPES) was selected as the starting material to fabricate the porous membrane substrate through the non-solvent induced phase inversion, and the membrane substrate was crosslinked by 1,3-propanediamine (PDA) and quaternized by trimethylamine (TMA) successively to fabricate the PES-based porous AEMs. After that, their structure and the DD-related performance were investigated in detail.

## 2. Materials and Methods

### 2.1. Chemicals and Materials

Chloromethylated polyethersulfone (CMPES) with a degree of chloromethyl substitution of 0.49 was kindly provided by Suzhou Tuopusi Environmental Protection Technology Co., Ltd. (Suzhou, China). N-methylpyrrolidone (NMP), 1,3-propanediamine (PDA) and trimethylamine (TMA) were purchased from Aladdin Chem., Co., Ltd. (Shanghai, China). Deionized water was used throughout the experiments.

### 2.2. Preparation of PT-PES Porous AEMs

CMPES porous membrane as substrate was simply fabricated via the phase inversion method [27]. The measured amount of CMPES polymer was dissolved in NMP under stirring to form a 25 wt% homogenous solution at room temperature, which was then cast onto the glass plate by a casting knife with a gap of 250 μm. After that, the glass was immediately dipped into a pure water bath to get the CMPES porous membrane substrate. The porous AEMs were prepared via a facile two-step modification of the obtained CMPES porous membrane substrate, which was simply immersed in a 1.5 mol L$^{-1}$ PDA solution at 40 °C for several hours (1–5 h) to conduct the crosslinking modification, and then immersed in a 1 mol L$^{-1}$ TMA solution at 60 °C for 12 h to ensure the entire quaternization

(See Figure 1). According to the immersing time of CMPES membrane substrate in PDA solution, the obtained AEMs were named PT-PES-Xh (X = 1, 2, 3, 4, 5).

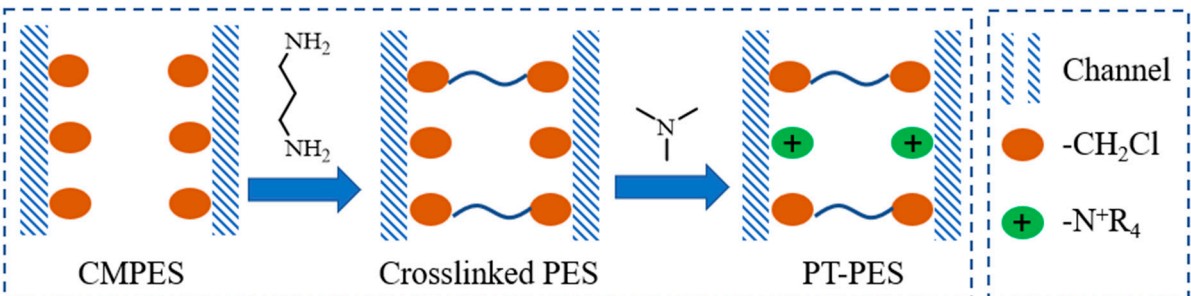

**Figure 1.** Schematic diagram of the preparation of PT-PES-Xh AEMs.

*2.3. Characterization*

2.3.1. Membrane Morphology, Composition and Thermal Stability

Scanning electron microscopy (SEM) analysis was taken out by a Nova Nano 400 scanning electron microscope (FEI, Hillsboro, USA) with an accelerating voltage of 10 kV, and X-ray photoelectron spectroscopy was conducted by an AXIS Nova spectrometer (Kratos Analytical Ltd., Manchester, UK) with a monochromated Al Ka source at a power of 150 W (10 kV 15 mA). Thermogravimetric analysis (TGA) was performed by a Mettler Toledo TGA/DSC thermal gravimetric analyzer (Zurich, Switzerland) at a heating rate of 10 °C min$^{-1}$ under pure argon.

2.3.2. Ion Exchange Capacity (IEC)

IEC test was performed via a method of ion exchange titration. Firstly, the dry weight ($W_{dry}$) of the AEM sample was measured, then the sample was soaked in a 0.2 mol L$^{-1}$ NaOH solution for 6 h at 25 °C to transform the chloride (Cl$^-$) form into the hydroxide (OH$^-$) form, followed by a thorough water wash, then the sample was ion-exchanged by immersing in a 0.5 mol L$^{-1}$ NaCl overnight. The mixture, composed of NaCl and the released NaOH solutions, was titrated by HCl solution, whose concentration ($C_{HCl}$) and consumption volume ($V_{HCl}$) were recorded to measure IEC of PT-PES AEMs as follows:

$$\text{IEC} = \frac{C_{HCl} \times V_{HCl}}{W_{dry}} \tag{1}$$

2.3.3. Water Uptake (WU)

The membrane sample was firstly immersed in water at 25 or 60 °C for 48 h; after that, the sample surface was quickly wipe-dried and the wet weight ($W_{wet}$) of the sample was obtained. The sample was then dried at 100 °C overnight prior to recording the dry weight ($W_{dry}$). WU of the sample can be measured as follows:

$$WU = \frac{W_{wet} - W_{dry}}{W_{dry}} \times 100\% \tag{2}$$

2.3.4. Diffusion Dialysis (DD)

DD runs were performed using a reported method [9]. Firstly, the mixture solution of HCl (0.2 mol L$^{-1}$) and FeCl$_2$ (1 mol L$^{-1}$) was made as the model feed solution, then the AEM sample with an effective area of 5.73 cm$^2$ was fixed between 2 chambers, which were denoted as feed side and water side, respectively. Afterwards, the feed side and water side were filled with 140 mL feed solution and 140 mL water, respectively. DD runs were then conducted by stirring these 2 chambers at 25 °C for 45 min. The concentrations of HCl and

FeCl$_2$ in two sides were measured by the titration with Na$_2$CO$_3$ solution (0.05 mol L$^{-1}$) and KMnO$_4$ solution (0.001 mol L$^{-1}$), respectively.

HCl dialysis coefficient ($U_{HCl}$) and FeCl$_2$ dialysis coefficient ($U_{FeCl_2}$) can be calculated as follows:

$$U = \frac{M}{At\Delta C} \tag{3}$$

where $M$ is the amount of HCl or FeCl$_2$ transported in moles, $A$ the effective membrane area in m$^2$, $t$ the time in h, and $\Delta C$ the logarithm average concentration of HCl or FeCl$_2$ between the two sides in mol m$^{-3}$ and defined as follows:

$$\Delta C = \frac{\left(C_f^0 - C_d^t - C_f^t\right)}{\ln\left[\left(C_f^0 - C_d^t\right)/C_f^t\right]} \tag{4}$$

where $C_f^0$ and $C_f^t$ are the feed concentrations of HCl or FeCl$_2$ at time 0 and t, respectively, and $C_d^t$ the dialysate concentration of HCl or FeCl$_2$ at time t. Considering the volume change caused by water transport, $\left(C_f^0 - C_d^t - C_f^t\right)$ is not equal to zero [28].

The acid/salt separation factor ($S_{HCl/FeCl_2}$) can be calculated as follows:

$$S_{HCl/FeCl_2} = U_{HCl}/U_{FeCl_2} \tag{5}$$

## 3. Results and Discussions

### 3.1. XPS

XPS analysis was conducted to evaluate the chemical compositions of CMPES membrane substrate and PT-PES-5h AEM. As shown in Figure 2, the survey spectra of CMPES and PT-PES-5h reveal five peaks at binding energies of 531.4 eV (O 1s), 399.6 eV (N 1s), 284.9 eV (C 1s), 199.9 eV (Cl 2p) and 168.1 eV (S 2p); moreover, the PT-PES-5h AEM is stronger in the peak density of N element and weaker in that of Cl element, as compared with CMPES substrate. It is identical to the nucleophilic substitution reaction between CMPES and functional amine agents including PDA and TMA, introducing amine groups into PT-PES AEMs accompanied by the exhaustion of -CH$_2$Cl groups from CMPES substrate. The high-resolution N 1s XPS spectra of CMPES substrate and PT-PES-5h AEM are shown in Figure 3. There is one peak at binding energy of 396.6 eV for CMPES substrate, while there are two peaks at binding energies of 399.6 and 402.3 eV for PT-PES-5h AEM. The newly formed peak (402.3 eV) is ascribed to the charged amine (NR$_4$$^+$) [29], proving the successful quaternization of CMPES substrate to prepare PT-PES AEMs.

### 3.2. SEM

SEM images of the CMPES substrate, PT-PES-1h AEM and PT-PES-5h AEM are shown in Figure 4. There is a typical asymmetrically porous microstructure for the CMPES substrate including a nano-porous top surface, a macro-porous supporting layer and a macro-porous bottom surface. The treatments of the PDA and TMA have a great influence upon the morphology of PT-PES AEMs. Compared with CMPES substrate, the pore size and density on the top surface of PT-PES-1h AEM are much lower, due to the crosslinking effect of PDA. From PT-PES-1h AEM to PT-PES-5h AEM, the pores on the top surface become unobservable with the further enhanced crosslinking effect. The morphology change on the bottom surface from CMPES substrate to PT-PES AEMs is similar to that on the top surface; in other words, with the increasing crosslinking time by PDA, the pore size and density firstly decrease and then the pores become unobservable. For the bottom surface, there is no significant change in the membrane morphology from CMPES substrate to PT-PES AEMs, indicating that PT-PES AEMs can maintain the porous microstructure, which is conductive to the ion transport because of the significantly enhanced free space volume and reduced ion transport resistance.

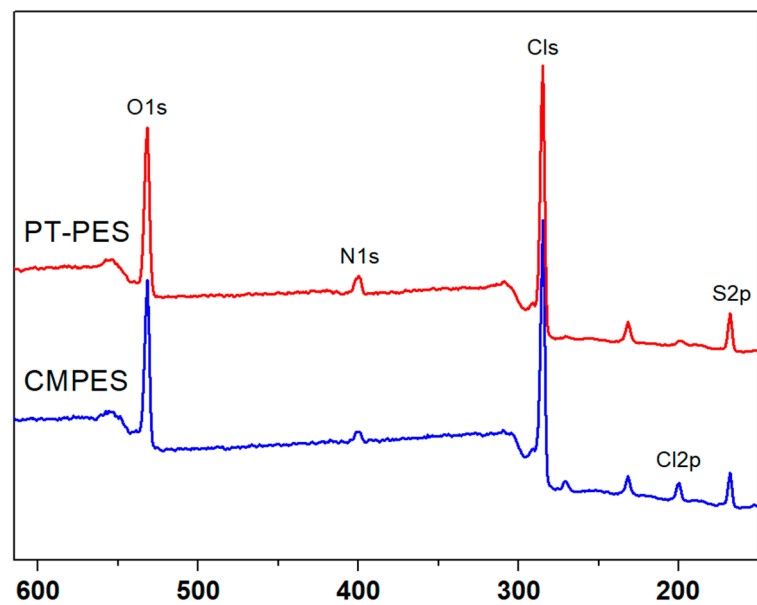

**Figure 2.** XPS survey spectra of the CMPES substrate and PT-PES-5h AEM.

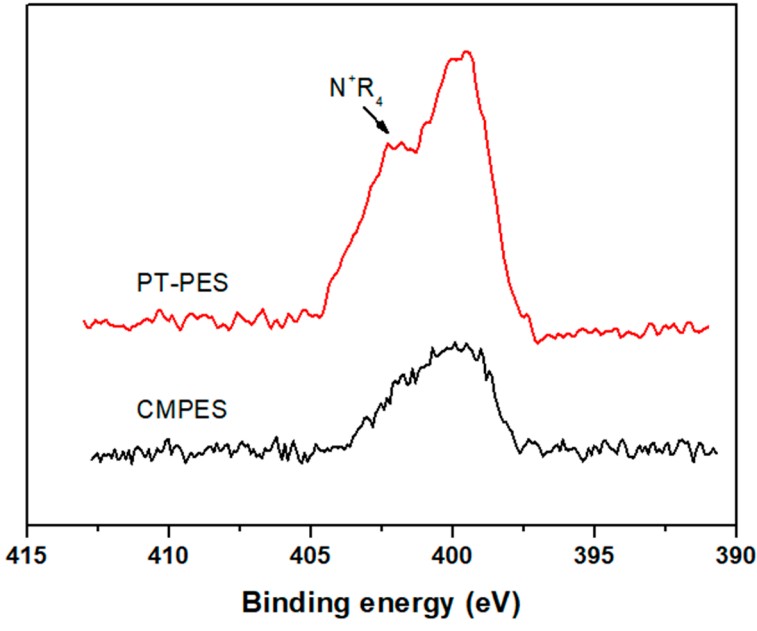

**Figure 3.** High-resolution XPS spectra of N1s for the CMPES substrate and PT-PES-5h AEM.

*3.3. Ion Exchange Capacity (IEC)*

As shown in Figure 5, the IEC value of PT-AEM-1h is 1.45 mmol g$^{-1}$; by contrast, the IEC value of the CMPES membrane substrate is almost 0 when tested in the preliminary experiment, due to the lack of anion exchange groups. From PT-PES-1h AEM to PT-PES-5h AEM with increasing immersion time in PDA solution, the IEC values decrease from 1.45 mmol g$^{-1}$ to 0.53 mmol g$^{-1}$. As mentioned above, CMPES substrate was firstly crosslinked by PDA and then quaternized by TMA. Considering the reaction site competition between PDA and TMA with -CH$_2$Cl groups of CMPES, the functionalization process of PDA would decide both the crosslinking and quaternization degrees of PT-PES AEMs. Therefore, there was less -CH$_2$Cl group left for the quaternization in CMPES substrate after a longer immersion time in PDA solution; consequently, the corresponding PT-PES AEM would have lower IEC.

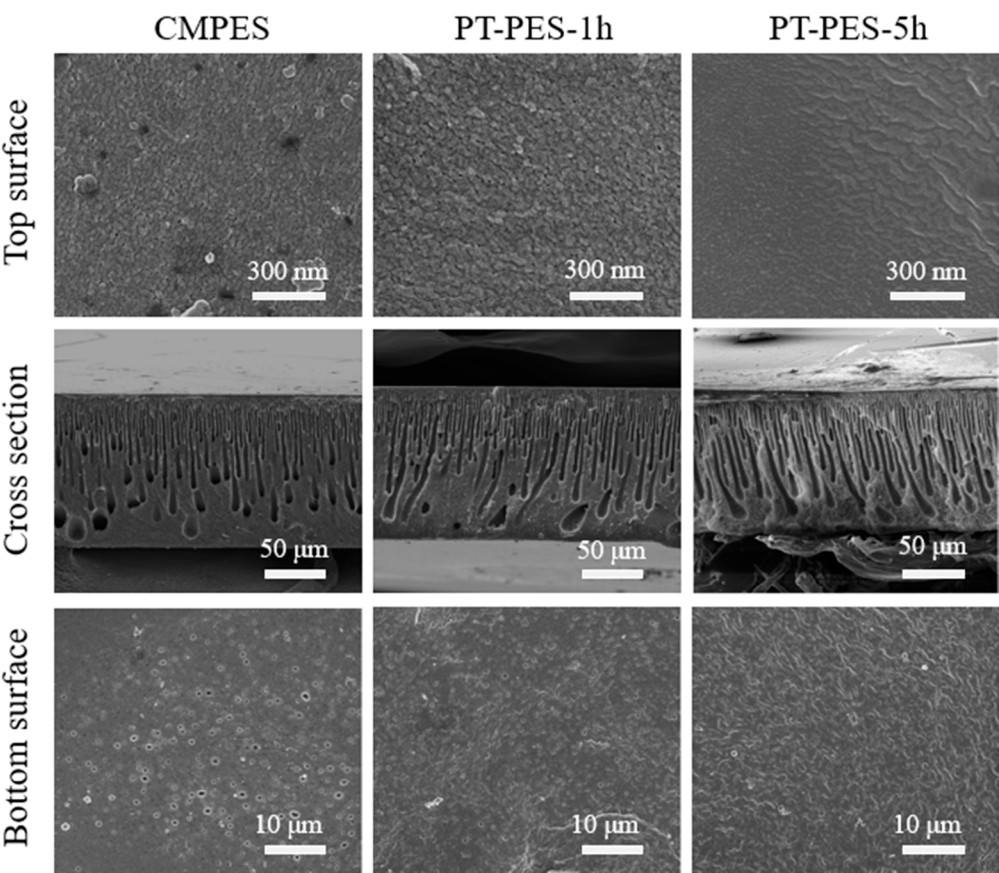

**Figure 4.** SEM images of CMPES substrate, PT-PES-1h AEM and PT-PES-5h AEM.

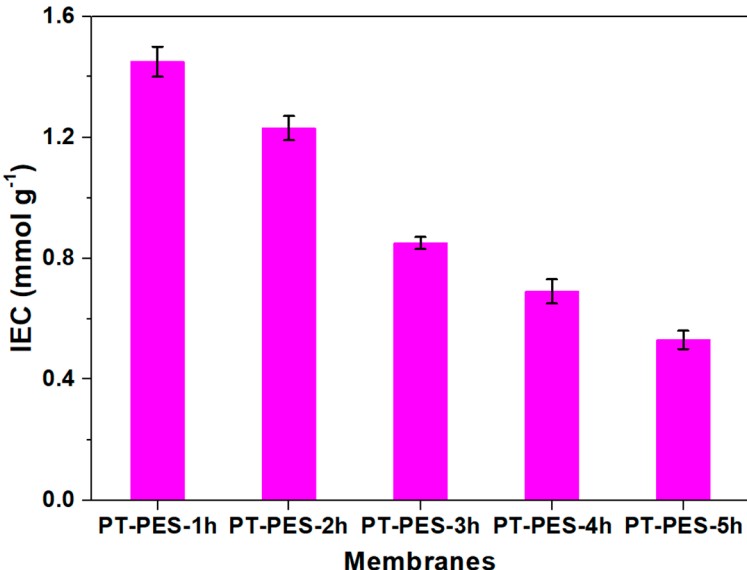

**Figure 5.** IEC values of PT-PES AEMs.

*3.4. Water Uptake (WU)*

WU values of the CMPES substrate and PT-PES AEMs at 25 and 60 °C are shown in Figure 6. As observed, WU value of the CMPES substrate is 150.3% at 25 °C. From PT-PES-1h AEM to PT-PES-5h AEM, WU values decrease from 350.4% to 200.4%. The reason is due to the decrease in the content of the hydrophilic quaternary ammonium groups (IEC), causing a decline in water absorption in PT-PES AEMs. Generally, WU value

of the membrane (e.g., CMPES substrate) at the higher temperature should be higher than that at low temperature, but it is worth remembering that PT-PES AEMs have similar WU values at both 25 and 60 °C. This means that the serious swelling at high temperature can be inhibited for PT-PES AEMs, attributed to the fully crosslinking effect by PDA as the crosslinker. Such phenomena prove the good swelling stability of PT-PES AEMs, and are consistent with our previous work [26].

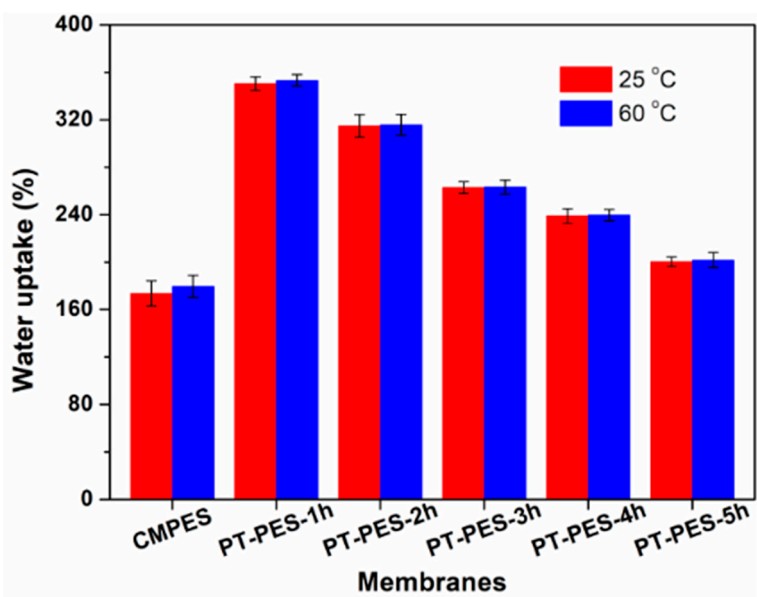

**Figure 6.** WU values of the CMPES substrate and PT-PES AEMs.

### 3.5. Diffusion Dialysis (DD) Performance

The DD performance of PT-PES AEMS including $U_{HCl}$ and $S_{HCl/FeCl_2}$ are represented in Figure 7. It is that $U_{HCl}$ value of CMPES substrate is almost 0 because of the absence of an anion exchange group [25], which is also consistent with the XPS and IEC results. From PT-PES-1h AEM to PT-PES-5h AEM, $U_{HCl}$ values of the PT-PES AEMs decrease from 0.056 m h$^{-1}$ to 0.026 m h$^{-1}$, while $S_{HCl/FeCl_2}$ values increase from 16.4 to 197.3. These phenomena can be explained by the physicochemical properties, especially the quaternization degree (IEC) and the membrane morphology. As discussed in the IEC section, from PT-PES-1h AEM to PT-PES-5h AEM, the quaternization degree (IEC) increased, resulting in an increase in ion transport as well as $U_{HCl}$. Moreover, the SEM results revealed that the pore size on the top surface would decrease with the increasing immersion time of PT-PES AEMs in PDA solution, significantly hindering the ion transport through the AEMs. Besides, the hindering effect on ion transport is more definite for Fe$^{2+}$ than H$^+$, thus leading to the increase in $S_{HCl/FeCl_2}$. In other words, the increase in $U_{HCl}$ is due to the increase in IEC, and the decrease in $S_{HCl/FeCl_2}$ is due to the enhanced compactness. In view of the balance of $U_{HCl}$ and $S_{HCl/FeCl_2}$, PT-PES-2h AEM with $U_{HCl}$ of 0.051 m h$^{-1}$ and $S_{HCl/FeCl_2}$ of 32.5 are the optimal ones among PT-PES AEMs, both of which are, respectively, 466.7% and 75.7% higher than the commercial DF-120 AEM, indicating the promising DD performance of our PT-PES AEMs for acid recovery. As compared with the other reported dense (symmetric) AEMs (See Table 1) [9–19], PT-PES-2h AEM also shows superior DD performance. The great enhancement in DD performance is attributed to the special microstructure of PT-PES AEMs. As discussed in the SEM section, PT-PES AEMs possess a high free space volume as compared with the common dense AEMs, and the ion transport resistance can be significantly reduced, according to the solution-diffusion mechanism [22,23].

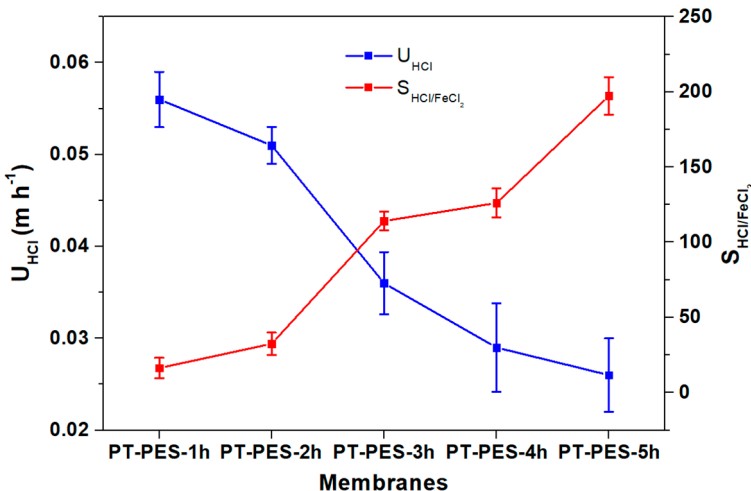

**Figure 7.** $U_{HCl}$ and $S_{HCl/FeCl_2}$ values of PT−PES AEMs for the acid recovery from the HCl/FeCl$_2$ mixture solution at 25 °C.

**Table 1.** $U_{HCl}$ and $S_{HCl/FeCl_2}$ values of the reported membranes at 25 °C using HCl/FeCl$_2$ solution as the model acidic waste solution.

| Membrane | Compactness | $U_{HCl}$ ($\times 10^{-3}$ m h$^{-1}$) | S | Ref. |
|---|---|---|---|---|
| PT-PES-2h | Porous | 51.0 | 32.5 | This work |
| DF-120 commercial membrane | Dense | 8.5 | 18.5 | [9] |
| Quaternized PPO-based hybrid membranes | Dense | 5.0–11.0 | 17.0–32.0 | [9] |
| PVA and glycidyl trimethyl ammonium chloride (EPTAC) blending membranes | Dense | 11.0–18.0 | 18.5–21.0 | [10] |
| PVA and multi-alkoxy silicon copolymer blending membranes | Dense | 10.0–17.0 | 24.0–30.1 | [11] |
| PVA treated with alkoxysilanes membranes | Dense | 8.0–10.0 | 15.9–21.0 | [12] |
| Quaternized poly (VBC-co-γ-MPS) membranes | Dense | 24.0–43.0 | 22.0–26.0 | [13] |
| Quaternized bionic multisilicon copolymers | Dense | 7.2–7.5 | 25.9–42.8 | [14] |
| Quaternized blending of PPO and PVA membranes | Dense | 21.0–49.0 | 26.0–39.0 | [15] |
| Quaternized aromatic amine-based hybrid PVA membranes | Dense | 17.2–25.2 | 14.0–21.0 | [16] |
| Quaternized PPO blending with PVA and silanol | Dense | 9.5–14.5 | 45.0–67.5 | [17] |
| Quaternized blending of PVC and P (DMAM-co-DVB) | Dense | 12.0–40.0 | 36–61 | [18] |
| Imidazolium functionalized hybrid membrane blending with PVA | Dense | 18.7–48.3 | 12.72–52.5 | [19] |

### 3.6. TGA

TGA was conducted for the investigation of the thermal stability of PT-PES AEMs. Figure 8 shows the TGA curve of PT-PES-5h AEM. As can be seen, there is a slight weight (~5%) loss when the temperature is below 100 °C, ascribing to the absorbed water evaporation, and can be ignored. Based on this, the starting degradation temperature of PT-PES AEMs when started is about 134 °C, attributed to the weight loss of the charged amine groups in PT-PES AEMs [30]. Since the operating temperature of the DD process is al-

ways below 80 °C, PT-PES AEMs therefore possess the desirable thermal stability for the practical application.

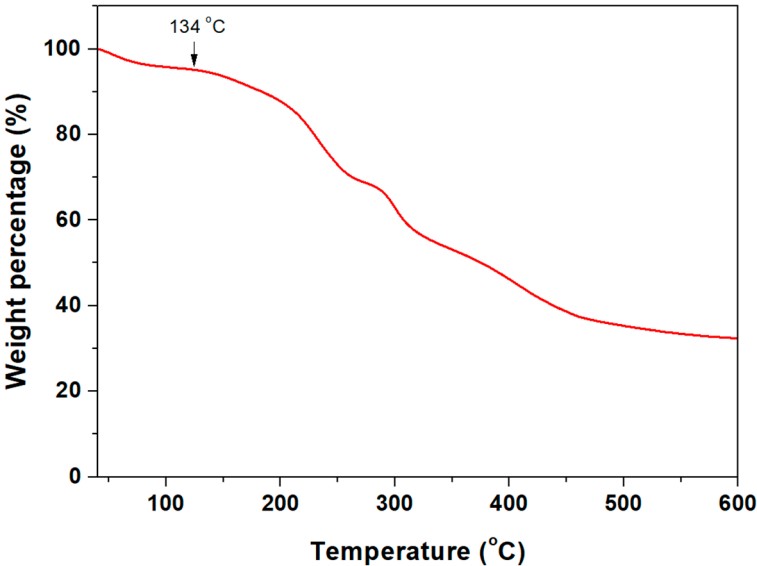

**Figure 8.** TGA curve of PT-PES-5h AEM.

## 4. Conclusions

Porous PES-based AEMs with high performance were fabricated for acid recovery via DD. Specifically, CMPES porous membrane substrate was successively crosslinked and quaternized by simple immersion in PDA and TMA solutions, in turn, to obtain the PT-PES AEMs, which showed promising DD performance as compared with the common dense AEMs due to the porous microstructure. Specifically, the optimal PT-PES-2h had $U_{HCl}$ of 0.051 m h$^{-1}$ and $S_{HCl/FeCl_2}$ of 32.5, both of which are 466.7% and 75.7% higher than the commercial DF-120 AEM, respectively. Therefore, PT-PES AEMs herein show great potential for the practical application in acid recovery by DD.

**Author Contributions:** Data curation, Writing—original draft, Formal analysis, Investigation, J.Y.; Data curation, Writing—original draft, Formal analysis, Investigation, G.D.; Formal analysis, Investigation, J.W.; Formal analysis, Investigation, S.P.; Formal analysis, Investigation, G.L.; Formal analysis, Investigation, X.S.; Writing—review and editing, D.T.; Writing—review and editing, J.C.; Conceptualization, Methodology, Resources, Funding acquisition, Supervision, Writing—review and editing, X.L. All authors have read and agreed to the published version of the manuscript.

**Funding:** This research was funded by the National Natural Science Foundation of China, grant number 21808038.

**Acknowledgments:** This work is supported by the National Natural Science Foundation of China (grant number 21808038) and the Award Program for Minjiang Scholar Professorship.

**Conflicts of Interest:** The authors declare no conflict of interest.

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
