# Peer review of "Porous Anion Exchange Membrane for Effective Acid Recovery by Diffusion Dialysis"

_processes, doi:10.3390/pr9061049_

Round 1

Reviewer 1 Report

Introduction: the authors propose the DF-120AEM as the benchmark, but no specification or further descriptions are presented. What are the major shortcomings of the process as today? What are the "conventional treatments" and what are their problems? The economic advance of DD is mentioned, if so, why are other methods being used? It seems to me that there is a potential economic advantage, if certain issues are resolved, but those are never presented in the introduction. Is DF120 asymmetric, symmetric???

The results presented are limited to the characterization and test of the prepared materials with a test solution. A comparison to other membrane (for example DF 120) already in the market is necessary .

PT PES 2h is presented as the optimal , but no discussion on how this conclusion as reached is presented. The authors claimed that this membranes provides a 466% and 76 % improvement over DF 120 properties, what is the source of the DF 120 properties?  Are those comparable, given that they were not measured alongside the new materials? 

TGA is shown for the PT PES 5h, while showing the TGA for the optimal material would make more sense. How does the degradation temperature change with time of crosslinking?

There are numerous language errors and typos throughout the manuscript. 

Author Response

Response to reviewer 1#‘s comments

Introduction: the authors propose the DF-120AEM as the benchmark, but no specification or further descriptions are presented. What are the major shortcomings of the process as today? What are the "conventional treatments" and what are their problems? The economic advance of DD is mentioned, if so, why are other methods being used? It seems to me that there is a potential economic advantage, if certain issues are resolved, but those are never presented in the introduction. Is DF120 asymmetric, symmetric???

Reply: Thanks for your valuable suggestion. The common methods for the treatment of the acidic waste water are neutralization, crystallization, thermal decomposition and solvent extraction, however, they suffer from the serious problems majorly including high energy consumption and further pollute by disposal of salts. Compared to these common methods of dealing with the acid waste, the diffusion dialysis (DD) process is known to be one of the most cost-effective technologies because of its unique features including lower energy consumption and lower installation and operating costs. Until now, DD technology is not yet well developed because of the major shortcomings especially the low processing capacity and efficiency, due to the low acid permeation of AEMs used. However, DD is still ecumenically promising considering the increasing cost for the treatment of the acidic wastewater by other common methods. This is the reason why the dialyzer using the famous commercial DF-120 AEM (dense and symmetric) developed by Tianwei Membrane Co. Ltd. (China) has achieved extensive industrialization, and the biggest goal of the research on DD is to develop an AEM with high acid dialysis coefficient, to improve the processing capacity and efficiency.

 The introduction section has been carefully revised.

The results presented are limited to the characterization and test of the prepared materials with a test solution. A comparison to other membrane (for example DF 120) already in the market is necessary.

Reply: Thank for your valuable suggestion. The DD performance of the reported dense AEMs have been added (See Table 1).

PT PES 2h is presented as the optimal, but no discussion on how this conclusion as reached is presented. The authors claimed that this membranes provides a 466% and 76 % improvement over DF 120 properties, what is the source of the DF 120 properties?  Are those comparable, given that they were not measured alongside the new materials? 

Reply: Thanks for your valuable suggestion. PT-PES-2h AEM was selected as the optimal one in view of the balance of and . The DD performance is cited from Luo’s work (Journal of Membrane Science 2010; 347:240-249).

TGA is shown for the PT PES 5h, while showing the TGA for the optimal material would make more sense. How does the degradation temperature change with time of crosslinking?

Reply: Thanks for your valuable suggestion.The thermal stabilities especially the starting degradation temperature of the porous AEMs for acid recovery are majorly related to the thermal stability of the quaternary ammonium groups, therefore, we chose PT-PES-5h for the TGA test.

There are numerous language errors and typos throughout the manuscript. 

Reply: Thanks for your valuable suggestion. The whole manuscript has been carefully revised.

Reviewer 2 Report

Please see the attached comments.

Author Response

Response to reviewer 2#'s comments

 The authors report acid recovery by diffusion dialysis using an in-house fabricated porous anion exchange membrane. The membrane shows superior performance (466.7% higher acid dialysis coefficient and 75.7% higher acid/salt separation factor) compared to commercial dense membranes. A membrane weight loss of ~5% in the region of operating temperature below 100 °C could be a concern. Overall, the membrane’s superior separation performance justifies further investigation into practical replacement of current dense membranes used in acid recovery diffusion dialysis.

Specific comments

Reply: The slight weight (~5%) loss when the temperature below 100 °C is ascribed to the absorbed water evaporation, and unrelated to the thermal stability, it is always ignored for the TGA analysis.

Line 44: “economic economically”, needs to be corrected

Reply: It has been corrected as “economically promising”

Line 35 and Line 46: a brief mention of the “conventional methods” or “common methods” would help the reader to understand the context and justification more.

Reply: Thanks for your valuable suggestion. The detail about the common methods has been added.

Line 51-58: can the performance indicators of the synthesised dense membranes mentioned be included as well. This can help to justify why the current research on porous membranes is necessary.

Reply: Thank for your valuable suggestion. The DD performance of the reported dense AEMs have been added (See Table 1).

Line 64-65: the “nano-64 porous selective layer and a macro-porous supporting layer” were made of which materials?

Reply: Thanks for your valuable suggestion, the whole membrane is made of CMPES with the grafted PMDETA.

Methods: Can more descriptive methods be added as supplementary information, to give, for example, information such as accelerating voltage in SEM analysis?

Reply: Thanks for your valuable suggestion, more detail has been added for the experimental section. The accelerating voltage in SEM analysis, the electron source and its power in XPS analysis, and the heating rate in TGA analysis have been added.

Figure 5: Needs to come after the text which describes it. Also, could the quality of this figure be improved?

Reply: Thanks for your valuable suggestion. All Figures are re-placed after the text. Fig. 5 and Fig. 7 with higher quality have been added.

Figure 7: Can results for commercial DF-120 AEM also be included to help with visual comparison?

Reply: Thanks for your valuable suggestion, the results of the commercial DF-120 AEM and the other reported dense AEMs has been added (See Table 1).

Reviewer 3 Report

This manuscript is well organized, and the authors obtained enough data to describe the experimental results. However, it has significant flaws provided below. It should be presented in the revised manuscript to be published.

The quality of some figures (especially Fig. 5 and 7) should be improved.

The authors must compare the performance (U and S) of the synthesized membrane in this study to other ones reported in the literature. Is it superior?

The authors should describe the relationship between the physical/chemical properties of the membranes synthesized under different conditions and their performances.

Author Response

Response to reviewer 3#

This manuscript is well organized, and the authors obtained enough data to describe the experimental results. However, it has significant flaws provided below. It should be presented in the revised manuscript to be published.

The quality of some figures (especially Fig. 5 and 7) should be improved.

Reply: Reply: Thanks for your valuable suggestion, Fig. 5 and Fig. 7 with higher quality have been added.

The authors must compare the performance (U and S) of the synthesized membrane in this study to other ones reported in the literature. Is it superior?

Reply: Thanks for your valuable suggestion. The comparison between our AEM and the reported AEMs has been taken, as shown in Table 1.

The authors should describe the relationship between the physical/chemical properties of the membranes synthesized under different conditions and their performances.

Reply: Thanks for your valuable suggestion. More detail has been added.

Round 2

Reviewer 3 Report

The manuscript was revised by reflecting some of my comments.
1) However, I did not get the answer to the question, "The authors should describe the relationship between the physical/chemical properties of the membranes synthesized under different conditions and their performances."
How was it revised?? I could not find any update on that. 
2) What is the contribution of the new author? I do not think that the revised manuscript was not updated with the help of the new author.

Author Response

The manuscript was revised by reflecting some of my comments.
1) However, I did not get the answer to the question, "The authors should describe the relationship between the physical/chemical properties of the membranes synthesized under different conditions and their performances."
How was it revised?? I could not find any update on that. 

Reply: The authors appreciate the reviewer's  rigor attitude,  the authors are sorry for the unsatisifed pre-reply. Now more detail has been added for the DD performance section. Specifically, the relationship between DD performance and the physicochemical properties especially IEC and membrane morphology has been discussed.
2) What is the contribution of the new author? I do not think that the revised manuscript was not updated with the help of the new author.

Reply: The authors are sorry for the carelessness. The two new authors make a distribution in the Writing - review & editing, we have revised the section of Author Contributions.